# Detecting sample swaps in diverse NGS data types using linkage disequilibrium

Nauman Javed[1,2], Yossi Farjoun [2], Tim J. Fennell[2], Charles B. Epstein [2], Bradley E. Bernstein [1,2] & Noam Shoresh [2✉]

As the number of genomics datasets grows rapidly, sample mislabeling has become a high stakes issue. We present CrosscheckFingerprints (Crosscheck), a tool for quantifying sample-relatedness and detecting incorrectly paired sequencing datasets from different donors. Crosscheck outperforms similar methods and is effective even when data are sparse or from different assays. Application of Crosscheck to 8851 ENCODE ChIP-, RNA-, and DNase-seq datasets enabled us to identify and correct dozens of mislabeled samples and ambiguous metadata annotations, representing ~1% of ENCODE datasets.

[1] Department of Pathology and Center for Cancer Research, Massachusetts General Hospital and Harvard Medical School, Boston, MA 02114, USA. [2] Broad Institute of MIT and Harvard, Cambridge, MA 02142, USA. ✉email: nshoresh@broadinstitute.org

Biomedical research is rapidly embracing large-scale analysis of next-generation sequencing (NGS) datasets, often by integrating data generated by consortia or many individual research labs. Parallelized NGS analysis of tissues from many different patients is also commonplace in clinical genomics pipelines. In these settings, sample or data mislabeling, where datasets are incorrectly associated with a donor, can lead to erroneous conclusions, misdirect future research, and affect treatment decisions[1–3] (Fig. 1a). Verifying the relatedness of samples that nominally share a

**Fig. 1 Incorporating linkage information allows robust comparison of sequencing datasets. a** Sample swaps and misannotations, where a sample is incorrectly attributed to the wrong donor, are a high stakes issue for large consortium projects and clinical science. **b** Our method compares reads from two datasets across a genome-wide set of linkage disequilibrium (LD) blocks (haplotype map). The single-nucleotide polymorphisms (SNPs) in each block are highly correlated with each other and have low correlation with SNPs in other blocks. Reads overlapping any of the SNPs in a given block inform the relatedness of the datasets, even when reads from the two datasets do not overlap one another. **c** Haplotype maps contain many large LD blocks. LD blocks are created using common, ancestry independent SNPs from 1000 Genomes. Most SNPs lie within blocks of size >2, which boosts the chances of reads to be informative. **d** Distribution of LOD (log-odds ratio) scores for 34,336 donor-mismatched (red) and 9767 donor-matched pairs (green) of public ChIP-, RNA-, and DNase-seq datasets from the ENCODE project. **e** LD-based method can correctly determine sample relatedness even at low sequencing coverage. Pairwise comparisons of reference dataset pairs at different subsampling percentages using two equally sized SNP panels—one panel contained only independent single SNPs, while the other contained only LD blocks. Donor-mismatched dataset pairs are colored red while donor-matched dataset pairs are green. **f** Comparison of NGSC and Crosscheck's classification of 34,336 donor-mismatched and 9767 donor-matched dataset pairs. Performance was measured in terms of the false flag rate (FFR), the fraction of donor-matched pairs incorrectly flagged as donor mismatches, and the false-match rate (FMR), the fraction of donor-mismatched pairs incorrectly identified as donor matches. Comparisons are classified as same-assay if the two datasets are from the same-assay type, and have the same target epitope in the case of ChIP-seq datasets. All other comparisons are classified as cross-assay. (Elements of (**a** and **b**) have been modified from a CDC publication (https://commons.wikimedia.org/wiki/File:Access_to_Health_Care-CDC_Vital_Signs-November_2010.pdf) which is under a CC BY-SA licence: https://creativecommons.org/licenses/by-sa/4.0/deed.en.)

donor is therefore a crucial quality-control step in any NGS pipeline.

Several methods utilize genetic information from NGS datasets as an endogenous barcode to verify sample relatedness[4–10]. The common logic behind these tools is that each genome harbors a unique set of single-nucleotide polymorphisms (SNPs) that are shared between datasets originating from the same donor. A limitation of these methods is their requirement that sequencing reads from both inputs overlap the exact genomic position of informative SNPs. When insufficient reads satisfy this condition—for example when the input datasets are shallow or target different genomic regions (i.e different transcription factors), the power to evaluate sample relatedness is compromised. Many NGS-based studies now integrate multiple types of assays[11–15] and utilize shallow sequencing to reduce cost at the expense of read depth. This is commonly encountered in highly multiplexed experiments, sequencing spike-ins, and large cohort sequencing efforts in population and cancer genomics (i.e. 1000 Genomes, structural variant calling). We therefore set out to develop a method for quantifying sample relatedness that was both robust to shallow sequencing depth and that could be systematically applied to modern large-scale projects incorporating multiple data types.

Linkage disequilibrium (LD) is the non-random association of alleles at different loci within a given population[16]. This association implies that comparing datasets across SNPs in high LD—termed LD blocks—would provide more statistical power to compare datasets than using single SNPs alone. Because of LD, two non-overlapping reads from different datasets may support (or provide evidence against) a common genetic background, as long as they overlap SNPs in the same LD block (Fig. 1b). For each input dataset, Crosscheck uses reads overlapping SNPs within each LD block to calculate a block allele fraction and compute diploid genotype likelihoods, which are then compared (Methods). The relative likelihood of a shared or distinct genetic background at each block is reported as a log-odds ratio (LOD score). These scores are combined across all blocks to report a genome-wide LOD score. This calculation relies on two approximations: that linkage between SNPs in an LD block is perfect and that SNPs in distinct blocks are independent. A positive LOD score indicates a higher likelihood that the two datasets share a donor, while a negative LOD score suggests that the datasets are from distinct donors. The Crosscheck calculation assumes that the two datasets are a priori equally likely to be from the same donor as they are from different ones. It is possible to incorporate a different prior expectation for a mismatch by shifting the LOD scores (Methods). Though the magnitude of the LOD score reflects genotyping confidence, simplifying assumptions prevent direct interpretation of the LOD score as a true likelihood ratio (Methods). Crosscheck is implemented as part of Picard-Tools (https://github.com/broadinstitute/picard), and is routinely used for quality control by the Broad Institute's Genomics Platform, using a small set of LD blocks optimized for use with whole-exome-sequencing data.

We reasoned that applying Crosscheck across a large, genome-wide set of LD blocks (haplotype map) would allow us to compare the genotype of diverse datasets and would be robust to low coverage and sequencing errors. We constructed a map consisting of nearly 60,000 common (minor allele frequency (MAF) ≥ 10%) bi-allelic SNPs from the 1000 Genomes[11] project, the majority of which lie in LD blocks of two or more SNPs in order to maximize the probability of informative read overlap (Fig. 1c, Methods). SNPs within each block are highly correlated ($r^2 > 0.85$), while SNPs between blocks are approximately independent ($r^2 < 0.10$). Increasing or decreasing the thresholds for within-block and between-blocks correlations by 0.05 had no effect on the method's

performance on a testing dataset (described in the next paragraph). Finally, in order to reduce bias from donor ancestry, we required that LD blocks have similar allele frequencies across different human sub-populations. The pipeline for creating haplotype maps exists as a standalone tool (https://github.com/naumanjaved/fingerprint_maps) and comes with pre-compiled haplotype maps for both hg19 and GRCh38. The pipeline can be customized to create LD blocks in specific genomic areas (i.e. coding regions) and with different parameters (i.e. different intra or inter-block $r^2$).

In the rest of this manuscript we demonstrate that Crosscheck, used with the haplotype map that we generated, can reliably detect donor mislabeling with fewer errors than other existing methods. It is particularly superior in challenging settings such as low sequencing depth or when comparing datasets from diverse data types. We demonstrate the suitability of Crosscheck for large-scale production operation by applying it to 8851 datasets from the ENCODE consortium, and discuss the misannotations that this analysis uncovered.

## Results

**Benchmarking**. To pilot our method, we calculated LOD scores between donor-matched and donor-mismatched pairs of public datasets from the ENCODE[12] database, which hosts data from thousands of diverse NGS experiments (Methods). Classification performance was measured in terms of the false flag rate (FFR), the fraction of donor-matched pairs incorrectly flagged as donor mismatches, and the false-match rate (FMR), the fraction of donor-mismatched pairs incorrectly identified as donor matches. Our testing set comprised all pairwise comparisons between 281 RNA-, DNase-, and ChIP-seq (targeting histones, CTCF, or POL2) datasets with verified donor annotations (Supplementary Data 1), and all donor-mismatched comparisons between 101 ChIP-seq experiments targeting transcription factors and chromatin modifiers (Supplementary Data 2). This resulted in a final testing set of 34,336 donor mismatches and 9767 donor matches. Regardless of the input assay or enrichment target, Crosscheck correctly classified almost all dataset pairs with 0% FMR and 0.01% FFR, and showed a clear separation between donor mismatches (negative LOD) and donor matches (positive LOD) (Fig. 1d). Our method therefore confidently detects donor-matched and donor-mismatched dataset pairs.

We next quantified how using LD blocks improves classification performance. We generated two equally sized subsets of our full haplotype map—one comprised solely of unlinked SNPs and the other containing only LD blocks with two or more SNPs, and used these to classify the same testing dataset pairs. To simulate sparse datasets generated by spike-ins and multiplexed sequencing, we conducted each comparison at a range of sequencing depths, expressed as the percentage of reads subsampled from the original datasets (Methods, Supplementary Fig. 1a). Using LD blocks significantly decreased FMR and FFR, particularly at lower read depths and for cross-assay/target comparisons (Fig. 1e, Supplementary Fig. 1b). For example, at 5% subsampling (≤~$10^7$ reads), using LD blocks decreased the FMR and FFR by nearly 10% relative to using single SNPs for cross-assay comparisons.

**Comparison with other methods**. As mentioned above, there are other tools that quantify genetic sample relatedness. For comparison purposes, we considered only methods that could be applied to the general use case that Crosscheck is designed to address, namely comparing any two NGS datasets, and that can be deployed at scale, so that calculating tens-to-hundreds of thousands of comparisons is tractable. Two of the methods we

examined, HYSIS[6] and BAM-matcher[7], did not satisfy these criteria. Two other tools, Conpair[8] and BAMixChecker[9], provided inconclusive results for a high percentage of the testing-set comparisons (Methods). NGSCheckmate[10] (NGSC) is a model-based method that compares datasets by correlating allele fractions across a panel of reference SNPs, and was the only other method that could be directly compared to Crosscheck on the testing dataset. At high and intermediate read depths, both methods show similar performance. At lower read depths (≤15% subsampling), however, Crosscheck outperforms NGSC, as indicated by a consistently lower FMR and FFR (Fig. 1f). Crosscheck is particularly effective at classifying cross-assay dataset pairs, where it shows a 2–3% lower FMR and FFR than NGSC at 5% subsampling. In these use cases, Crosscheck performs better than NGSC due to its use of LD and the large number of SNPs in the haplotype map. Using LD blocks allows comparison of non-overlapping reads, while using a large set of SNPs increases the chance that input datasets will contain genetically informative reads. An illustrative example is a specific comparison between two ChIP-seq datasets, one targeting H3K27me3 and the other H3K27ac. At 5% subsampling, these datasets cover 8% and 2% of the genome, respectively, and overlap at only 0.02%, which is expected from these mutually exclusive histone modifications. Given this small set of potentially informative reads, NGSCheckmate wrongly concludes that the datasets are derived from the same donor, while Crosscheck is still able to make the correct call (Supplementary Fig. 1e). We have also tested Crosscheck, NGSC, BAMixChecker, and Conpair on sample pairs from seven donors that are genetically related. We found that Crosscheck can identify all pairs of samples from related individuals as donor mismatches, and is superior in this context to the other tools (Supplementary Fig. 2).

Finally, we used the distribution of LOD scores from incorrectly classified pairs to define an inconclusive LOD score range of $-5 < \text{LOD} < 5$, in which a dataset pair cannot be confidently classified (Methods, Supplementary Fig. 1c). Outside of this range, any pair with LOD ≥ 5 is denoted a donor match, and those with LOD ≤ −5 are flagged as donor mismatches. The inconclusive range highlights the interpretability of Crosscheck's LOD score relative to NGSC's binary outputs (match or mismatch), since clear donor mismatches can be prioritized and investigated separately from inconclusive comparisons. We conclude that using Crosscheck with a full haplotype map enables more accurate detection of donor-mismatched pairs in diverse and shallow collections of data.

**Crosscheck analysis of ENCODE data.** To illustrate the utility of our method on a consortium-scale dataset, we next analyzed the remaining datasets in ENCODE. We used our method to verify the donor annotation for all human hg19 aligned DNase-, RNA-, and ChIP-seq datasets in the ENCODE database whose annotated donor was represented by at least 4 datasets—a total of 8851 datasets (Fig. 2a, Supplementary Data 3). To scale our analysis to a database of this size, we compared each dataset to a set of three representative datasets from its annotated donor, and flagged any dataset with LOD < 5 for further review (Methods). To exclude the possibility that the representative set for each donor contained a donor mismatch, we required that all pairwise comparisons between representative datasets yield an LOD score ≥5. This strategy scales linearly with the size of the database, and in our case results in a 1000-fold reduction in computation relative to performing all pairwise comparisons.

Our strategy confirmed the annotated donor for 97% of datasets. The remaining 3% (256 datasets) were flagged as potential donor mismatches (LOD ≤ −5), and only ~0.1% yielded

inconclusive results (−5 < LOD < 5) (Fig. 2b). We next compared each flagged mismatch to the representative datasets for each of the ENCODE donors in order to nominate a true donor identity. We also compared each flagged mismatch to all other flagged mismatches in order to identify genetically consistent clusters and uncover patterns of mislabeling.

This analysis uncovered three major categories of mislabeling (as well as a small fraction, 0.4%, of datasets that exhibited a pattern consistent with cross-sample contamination, as described in Methods and Supplementary Fig. 3). The first is a straightforward error where cells from one donor are mistakenly labeled as deriving from a different donor. The likelihood of such a mistake increases when working with several cell lines that are each used in a large number of experiments. For example, out of four flagged datasets labeled as K562, two were shown to actually derive from GM12878 cells while the other two derived from HEK293 cells. This type of mislabeling may also occur for primary cells or tissues when many biological samples from multiple donors are obtained from the same source, as in the case of 300 embryonic tissue samples processed by ENCODE from a single lab.

The second class of mislabeling occurs when biological samples of the same cell type from multiple donors are incorrectly labeled as deriving from a single donor. This is the case with some of the commercially available primary cell lines that have been deeply interrogated by the consortium over more than a decade, and for which cells have been procured multiple times. For example, HUVEC cells are annotated as being derived from two different donors in the ENCODE metadata. However, our analysis indicates that HUVEC samples actually derive from at least five distinct donors (Fig. 2c). This mis-annotation went undetected by ENCODE's previous quality control pipelines because all samples were of the same cell type and so exhibited similar epigenetic profiles.

The HUVEC example also highlights the third type of labeling inaccuracy, in which a single donor is accessioned multiple times by dozens of different labs over several years. This results in slight variations in donor name or description, leading to genetically identical samples being incorrectly attributed to distinct donors. For example, some samples deriving from putative donor A are attributed to HUVEC donor 1, while other samples from donor A are attributed to the distinct HUVEC donor 2.

Overall, our analysis of the ENCODE dataset suggested that substantive mislabeling error occurred at a rate of ~1%. For these datasets, true donor identities were confirmed using ENCODE's extensive metadata records and all mislabeled datasets were corrected (Methods).

In conclusion, we present a robust and easy-to-use method for quantifying sample relatedness which outperforms similar methods. Combined with our method for database analysis and haplotype map, CrosscheckFingerprints can be readily applied for detecting sample mislabeling in large, diverse databases without any optimization. We suggest it as a critical component of any NGS quality control pipeline.

## Methods

**LOD derivation.** Here, a basic overview of the fingerprinting LOD score derivation is provided. A more detailed derivation is available at the Picard repository at: https://github.com/broadinstitute/picard/raw/master/docs/fingerprinting/main.pdf

Consider an LD block/locus containing a single bi-allelic SNP with major allele $A$ and minor allele $B$, and two sequencing datasets $x$ and $y$. Let $\theta$ and $\varphi$ denote the diploid haplotype of datasets $x$ and $y$, respectively, at this locus. $\theta$ and $\varphi$ can each take one of three possible haplotypes: $AA$, $AB$, or $BB$. Let $s$ be a Bernoulli random variable where $s = 1$ denotes a sample swap (indicating that $x$ and $y$ arose from two independent individuals) with posterior probability $p(s = 1 \mid x, y)$, and $s = 0$ denotes a shared genetic origin (the samples came from the same individual). Using Bayes' rule and the prior probability of no-swap, the posterior odds ratio of a

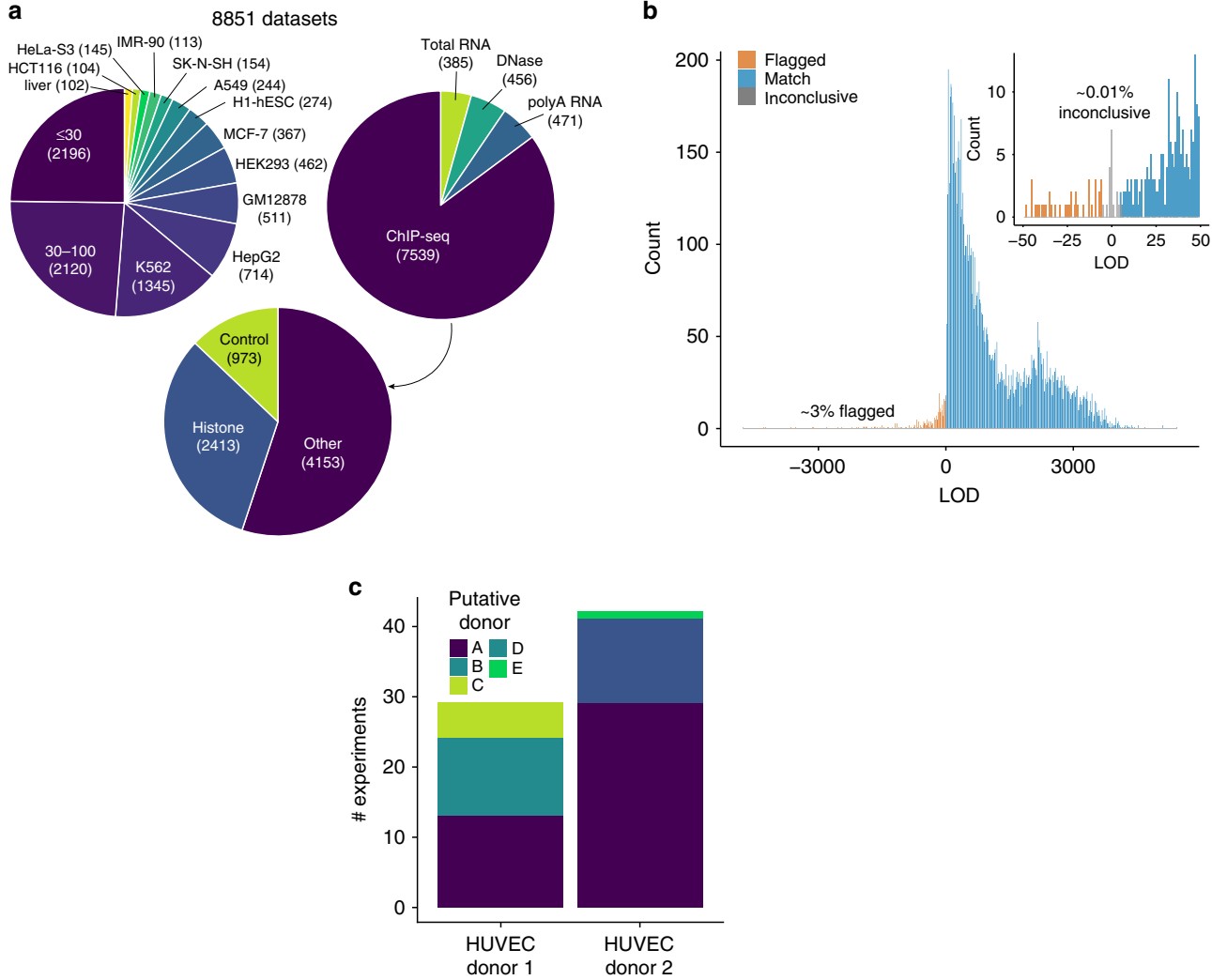

**Fig. 2 Overview of ENCODE database swap detection. a** Overview of 8851 genotyped datasets from ENCODE, partitioned by cell type (top left), assay type (top right), and by target for ChIP-seq (bottom). Cell types that had less than 100 datasets derived from them were pooled—so all the datasets from them are grouped into those with less than 30 datasets or those with 30-100 datasets. All hg19 aligned reads from total RNA-, polyA RNA-, ChIP-, and DNase-seq experiments performed on samples belonging to donors with at least four datasets in total were included in the analysis. All ChIP-seq targets, including histone modifications (HM), transcription factors (TF), chromatin modifiers (CM), CTCF, and control experiments were included. **b** Distribution of LOD scores from ENCODE genotyping. Each dataset was compared to three representative datasets from its nominal donor. Any dataset scoring negatively against any of the three representatives was flagged for further review. A comparison resulting in an LOD score between −5 and 5 was deemed inconclusive (insufficient evidence to indicate shared or distinct genetic origin). **c** Each flagged sample was compared to all other samples from its nominal donor, as well as the representatives for all other donors in our database to nominate true donor identity and identify genetically consistent sub-clusters. Comparisons of flagged samples between two HUVEC donors reveal five genetically distinct clusters.

no-swap vs. swap is given by

$$\frac{p(s=0 \mid x, y)}{p(s=1 \mid x, y)} = \frac{p(x, y \mid s=0)}{p(x, y \mid s=1)} \frac{p(s=0)}{p(s=1)}. \quad (1)$$

We assume that in the case of a swap, the distinct individuals are independently sampled from the population and that samples from the same individual have the same genotype, allowing us to write $p(\theta, \varphi \mid s) = p(\theta)p(\varphi)$ for $s=1$ and $p(\theta, \varphi \mid s) = p(\theta)$ if $\theta = \varphi$. Given that $x$ is conditionally independent of $\varphi$ and $y$ given $\theta$, and $y$ is conditionally independent of $\theta$ given $\varphi$, we can also write $p(x, y \mid \theta, \varphi) = p(x \mid \theta)p(y \mid \varphi)$.

With these two expressions, we derive that

$$p(x, y \mid s) = \sum_{\theta, \varphi} p(x, y \mid \theta, \varphi, s)p(\theta, \varphi \mid s)$$

$$= \begin{cases} \sum_{\theta} p(x \mid \theta)p(\theta) \sum_{\varphi} p(y \mid \varphi)p(\varphi) & \text{if } s=1 \\ \sum_{\theta=\varphi} p(x \mid \theta)p(y \mid \varphi)p(\theta) & \text{if } s=0 \end{cases}. \quad (2)$$

Substituting the results of Eq. (2) into Eq. (1), we rewrite the posterior odds of no-swap as

$$\frac{\sum_{\theta=\varphi} p(x \mid \theta)p(y \mid \varphi)p(\theta)}{\sum_{\theta} p(x \mid \theta)p(\theta) \sum_{\varphi} p(y \mid \varphi)p(\varphi)} \cdot \frac{p(s=0)}{p(s=1)}. \quad (3)$$

Next, we consider evidence over multiple blocks $i$ with correspondingly indexed $\theta_i$, $\varphi_i$, $x_i$, and $y_i$. We assume that the haplotypes at distinct blocks are independent, and that reads at one block give no information about another. In practice, this assumption is enforced by guaranteeing that a single read cannot be used to provide genotype evidence at more than one locus. We calculate $p(x \mid \theta) = \prod_i p(x_i \mid \theta_i)$ and $p(y \mid \varphi) = \prod_i p(y_i \mid \varphi_i)$, and substitute into Eq. (3) to get

$$\prod_i \left( \frac{\sum_{\theta_i=\varphi_i} p(x_i \mid \theta_i)p(y_i \mid \varphi_i)p(\theta_i)}{\sum_{\theta_i} p(x_i \mid \theta_i)p(\theta_i) \sum_{\varphi_i} p(y_i \mid \varphi_i)p(\varphi_i)} \right) \cdot \frac{p(s=0)}{p(s=1)}. \quad (4)$$

Finally, since the odds ratio of no-swap to swap may vary by several orders of magnitude depending on the input files, we compute the base 10 logarithm in order

to facilitate comparison and interpretation:

$$\text{LOD} = \log\left(\frac{\text{odds}_{\text{same individual}}}{\text{odds}_{\text{different individual}}}\right)$$

$$= \sum_i \log\left(\frac{\sum_{\theta_i=\varphi_i} p(x_i|\theta_i)p(y_i|\varphi_i)p(\theta_i)}{\sum_{\theta_i} p(x_i|\theta_i)p(\theta_i) \sum_{\varphi_i} p(y_i|\varphi_i)p(\varphi_i)} \cdot \frac{p(s=0)}{p(s=1)}\right). \quad (5)$$

The program assumes a conservative prior of $\frac{p(s=0)}{p(s=1)} = 1$ by default. A different prior would result in a shift of the LOD score by a constant, and users may adjust the LOD score by such a constant as needed on a case-by-case basis. A positive LOD (log-odds ratio) is interpreted as evidence for the two datasets $x$ and $y$ arising from the same individual, while a negative LOD is evidence of a sample swap, i.e., the two datasets arose from different individuals. Scores close to zero are inconclusive, and tend to result from low coverage, or poor overlap between the two datasets, at the observed sites.

To see the expected maximal contribution of a single locus, we assume that the likelihoods in (5) are vanishingly small when the data does not match the genotype. Thus, the LOD for a single locus reduces to $-\log p(\theta)$. The expected LOD contribution needs to be marginalized over the different possible genotypes, leading to a $-\sum_\theta p(\theta)\log p(\theta)$, which obtains a maximal value of $1.5\log_{10} 2 \approx 0.45$ at an allele frequency of 0.5 (leading to $p(\theta = AA) = 0.25$, $p(\theta = AB) = 0.5$, and $p(\theta = BB) = 0.25$). This means that when creating the haplotype map, it is most informative to choose variants with an allele frequency close to 0.5.

There is no theoretical lower limit to the contribution of a single locus. This is because, in theory, overwhelming evidence (hundreds of genetically consistent, high-quality reads) of different genotypes for two datasets at even a single locus is sufficient to rule out that the samples are derived from the same donor. However, as noted below in the section on the limitations of LOD calculation, there are multiple factors that this formulation does not account for. Our approach ultimately relies on cumulative evidence, albeit noisy, from a large number of loci, rather than looking for the small number of high-confidence cases. It is for this reason that in the implementation of Eq. (5) in the code, we have included an explicit lower cap on the possible contribution of any single LD block. The selection of the specific value at which to cap the negative contribution was guided by the following argument: We consider a single specific locus, and assume a conservative prior, $\frac{p(s=0)}{p(s=1)} = 1$. In addition, we assume that at that locus one dataset is only compatible with a single genotype, namely $p(y|\theta)$ is nonzero for only one value of $\theta$. In this case the contribution to the likelihood ratio for that locus reduces to

$$\frac{p(x\mid\theta)p(y\mid\theta)p(\theta)}{\left(\sum_{\theta_i} p(x\mid\theta_i)p(\theta_i)\right)p(y\mid\theta)p(\theta)} \gtrsim p(x\mid\theta). \quad (6)$$

If both samples are in fact from the same donor, and the discrepancy between $x$ and $\theta$ is due to a sequencing error, $10^{-3}$ is a reasonable ballpark estimate of $p(x\mid\theta)$[17]. With this, the actual score calculated by Crosscheck is

$$\text{LOD}' = \sum_i \max\left(\log\left(\frac{\sum_{\theta_i=\varphi_i} p(x_i\mid\theta_i)p(y_i\mid\varphi_i)p(\theta_i)}{\sum_{\theta_i} p(x_i\mid\theta_i)p(\theta_i)\sum_{\varphi_i} p(y_i\mid\varphi_i)p(\varphi_i)} \cdot \frac{p(s=0)}{p(s=1)}\right), \sigma\right), \quad (7)$$

where $\sigma = -3$ by default, and is a parameter that can be set by the user.

**Calculation of data likelihoods p(x | θ) from sequencing reads**. The program assumes that sequencing data arrive in the form of reads from a single individual (i.e. not contaminated), from a diploid location in the genome, and with no reference bias. Only non-secondary, non-duplicate reads with mapping quality greater than 20 are used to calculate likelihoods. In addition, bases must have a quality score of at least 20 and must agree with either the reference or pre-determined alternate base to support observations at haplotype blocks. Since the algorithm assumes that read evidence is independent, the reads should have been duplicate marked prior to fingerprinting. The algorithm does not use SNPs from the same read-pair twice, since this would violate the assumption of independence.

Consider a dataset $x$ for which we observe $n$ total sequencing reads, denoted by $r_k$, at a locus containing a single bi-allelic SNP with major allele $A$ and minor allele $B$. The possible block haplotypes are then $\theta \in \{AA, AB, BB\}$. For each read $r_k$ which overlaps the SNP, let $o_k \in \{A, B\}$ denote the observed SNP allele and let $e_k \in (0, 1)$ denote the probability of error of each observation (the quality score). We seek to compute the likelihood of the data (the sequencing reads $r_k$) given the haplotypes. The likelihood of a single base observation $p(o_k, e_i \mid \theta)$ is expressed by

$$p(o_k, e_k \mid \theta) = \begin{cases} I_B(o_k)e_k + I_A(o_k)(1 - e_k) & \theta = AA \\ 0.5 & \theta = AB \\ I_A(o_k)e_k + I_B(o_k)(1 - e_k) & \theta = BB, \end{cases} \quad (8)$$

where $I$ is an indicator function such that $I_A(o) = \begin{cases} 1 \text{ if } o = A \\ 0 \text{ if } o = B \end{cases}$ and $I_B(o) = \begin{cases} 1 \text{ if } o = B \\ 0 \text{ if } o = A \end{cases}$ and the assumption is that an error will cause a switch in the observed allele from $A$ to $B$.

The likelihood model for all reads $r$ can then be written as

$$p(r \mid \theta) = p(o, e \mid \theta) = \prod_{k=0}^{n} p(o_k, e_k \mid \theta). \quad (9)$$

**Incorporation of linkage information**. The calculations above assume an LD block containing a single SNP for ease of computation, but the framework is easily extended to account for LD blocks containing multiple SNPs, which increases power of comparison. Each LD block used for genotyping contains an "anchor" SNP which is in high linkage with all other SNPs within the block, and independent of all other anchor SNPs in other blocks. Given that all SNPs in a block are tightly linked (enforced with a strict $r^2$ correlation cutoff), we make the simplifying assumption that the genotype at any SNP within an LD block is perfectly correlated with the genotype of the anchor SNP, and that all SNPs within a block have the same allele frequency, equal to that of the anchor SNP. Then, reads overlapping any SNP within a block can be used to infer a total block haplotype, which is represented by the possible diploid genotypes of the anchor SNP. For example, consider an anchor SNPs $S_1$ with major allele $A$ and minor allele $B$, and a linked SNP $S_2$ with major allele $C$ and minor allele $D$. Then any observation of allele $C$ at SNP $S_2$ is taken as evidence of allele $A$ at $S_1$, and any observations of allele $D$ at $S_2$ is taken as evidence of allele $B$ at $S_1$. Using this strategy, evidence across all SNPs within a block can be used to infer a total block haplotype, which can be represented by the three possible diploid genotypes of the anchor SNP. That is, for an anchor SNP with major allele $A$ and minor allele $B$, the possible block haplotypes are $AA$, $AB$, and $BB$, with prior probabilities dependent on the allele frequencies of $A$ and $B$.

**Limitations of LOD calculation**. Though the magnitude of the LOD score reflects greater genotyping confidence, it cannot be directly interpreted as a likelihood ratio (e.g. an LOD of 200 does not correspond to a $10^{200}$ probability of a shared vs. different genetic origin), as the model does not fully account for sequencing noise, data quality, contamination, and relatedness. In addition, we did not model the incomplete dependence between haplotype blocks, nor the incomplete dependence of SNPs within blocks.

Our framework also assumes that the only two sources of a base are the observed allele or a sequencing error. This assumption can lead to incorrect results in the cases where a sample has particularly noisy data due to pre-sequencing events (such as PCR or FFPE processing), non-conforming LD blocks, or high contamination. These samples could be genotyped as heterozygous due to the noisy region or the non-confirming LD block structure. Including these error modes into the model would increase robustness and accuracy.

**Implementation details**. Crosscheck is implemented as part of the Picard-Tools suite, a set of Java command line tools for manipulating high-throughput sequencing data. It accepts VCF/BAM/SAM formatted inputs and can perform comparisons at the level of samples, libraries, read-groups, or files. Crosscheck is provided alongside a utility called ExtractFingerprints which for an input bam, outputs a VCF containing the genotypes and genotype likelihoods across all LD blocks within the supplied haplotype map. This VCF can be used to store fingerprints for downstream analyses or for use with Crosscheck. More information is available at https://github.com/broadinstitute/picard.

**Runtime and memory requirements**. For BAM mode, running Crosscheck requires approximately 2.5 gb RAM for a single input pair of BAMs. Runtime is dependent on the size of the input file. Based on our benchmarking experiments, runtimes are <10 min for DNAse-seq, <30 s for ChIP-seq, and are on average about 2 h for RNA-seq datasets. For VCF mode, Crosscheck requires approximately 2.5 gb of ram for a single pair of inputs, with runtimes <30 s using the standard hg19 haplotype map. CrosscheckFingerprints is multi-threading enabled in order to speed up comparisons and fingerprint generation when multiple input pairs are provided. All comparisons were conducted on Intel(R) Xeon(R) CPU E5-2680 v2 @ 2.80 GHz processors.

**Map construction overview**. Maps are constructed from 1000 Genomes[11] phase 3 (1000GP3) SNPs which are bi-allelic, phased, and have an MAF ≥10%. This MAF threshold is introduced since the expected maximal LOD contribution is obtained at an allele frequency of 0.50 (intuitively, rare variants are unlikely to be present in either of two samples being compared from different individuals). Additionally, SNPs must not differ in their MAF by more than 10% between the five ancestral sub-populations (AFR, SAS, EAS, EUR, AMR) present in 1000GP3. This is to correct for potential sub-population bias due to differing linkage and MAF frequency of SNPs across different populations. Using PLINK2 (ref. [18]), we pruned SNPs meeting these criteria in order to create an independent set of "anchor" SNPs, between which no pairwise $r^2$ correlation exceeded a threshold of 0.10. A window size of 10 kilobases (kb) and a slide of 5 SNPs were used for pruning. By creating this set of independent SNPs, we ensure that individual haplotype blocks are independent from each other. Next, we greedily added SNPs to the blocks represented by the anchor SNPs. Adding was done in order of LDScore[19] of the anchor SNPs, with the highest LDScoring anchor SNP first (LDScore is the sum for the $r^2$

correlations of each SNP with all other SNPs within a 1 cM window on either side). Recombination maps containing mappings between genomic coordinates and recombination rates for both the hg19 and GRCh38 assemblies were obtained from http://bochet.gcc.biostat.washington.edu/beagle/genetic_maps/ and http://mathgen.stats.ox.ac.uk/impute/1000GP_Phase3/. We only added SNPs if their correlation with the anchor SNP has $r^2 \geq 0.85$ and they were located within a genomic window of 10,000 kb. In this way, we prioritize the creation of larger, more genetically informative blocks that span several kb regions. The haplotype maps used for the ENCODE database analysis and benchmarking, along with the python code used to generate them, are available at: https://github.com/naumanjaved/fingerprint_maps.

**Constructing maps only containing LD blocks or single SNPs**. The map containing only single SNP blocks was constructed by aggregating all SNPs in the full haplotype map not in strong linkage ($r^2 \geq 0.85$) to other SNPs, resulting in 20,792 SNPs. To construct the map containing only blocks with size ≥2 used to quantify the benefits of accounting for linkage, we subsampled the full haplotype map. Starting with the largest blocks by number of SNPs, blocks were successively added to this map until the total number of SNPs approximately reached the number of SNPs in the map containing only independent SNPs (20,801).

**Testing-set construction**. *281 ChIP-seq, RNA-seq, and DNase-seq datasets with ground-truth annotation:* To create a testing set of files to evaluate our method's performance and benchmark it against other tools, we downloaded 281 hg19 bams from RNA-seq, DNase-seq, and ChIP-seq (targeting histone modifications, CTCF, or POL2) from the ENCODE Tissue Expression (ENTEX) project. The ENTEX project contains datasets from experiments on samples derived from four different tissue donors, each of which has whole-genome sequencing (WGS) data available. The WGS data for each donor can be used to verify the nominal donor of each dataset comprising the testing set. For each dataset, the corresponding hg38 alignments were compared against the hg38 WGS alignments for its nominal donor. Only datasets that yielded a positive LOD score >5 using CrosscheckFingerprints (with the full hg38 version of haplotype map) and a "match" result from NGSCheckMate were included in the testing set. The final testing set of files and accompanying metadata are included in Supplementary Data 1.

*101 transcription factor and chromatin modifier (CM) ChIP-seq datasets without ground-truth annotation:* To test Crosscheck and other methods on transcription factor and chromatin modifier datasets, we downloaded 101 hg19 ChIP-seq datasets from the ENCODE project. For these datasets, there was no ground-truth donor sequencing data available for the nominal donor as there was for the ENTEX datasets. In this case, the false-mismatch rate (incorrect genotyping call for a donor-matched pair) cannot be assessed, since there is a non-negligible probability that one of the two datasets with the same nominal donor annotation is incorrectly annotated. However, the FMR can still be assessed, since we estimate that the probability that two datasets with different donor annotations may actually share the same true donor is very low. Therefore, we only characterized the ability of NGSCheckmate and Crosscheckfingerprints to accurately classify donor-unmatched pairs for this testing set. In the context of detecting sample swaps, this performance measure is also more relevant than the accurate detection of donor-matched datasets. All datasets and accompanying metadata are available in Supplementary Data 2.

**BAM pre-processing and downsampling for benchmarking experiments**. Datasets were sorted using Samtools[20] and processed using Picard's MarkDuplicates tool with default settings to remove duplicates. We noted that collapsing duplicates was especially important for RNA-seq datasets since PCR bias can alter allele fractions and lead to incorrect sample classification. Downsampling was conducted on the duplicate marked, sorted files using the command *samtools view –s seed.F* with a seed value of 5.

**Benchmarking with NGSC and Crosscheck**. To speed up analysis of a large number of bams with NGSCheckmate, we followed the author recommendations[10] and created VCFs for each input file using the default provided SNP panel from the NGSCheckMate github and the command *samtools mpileup-I -uf hg19.fasta -l SNP_GRCh37_hg19_woChr.bed sample.bam | bcftools call -c - >./sample.vcf*. NGSC was then run in batch mode using default settings with the hg19 reference SNP panel. For Crosscheck, we first used Picard's ExtractFingerprint utility with default settings and the standard hg19 haplotype map to pre-compute VCFs for each input bam. Comparisons were then conducted using Crosscheck's batch mode with default settings and the standard hg19 map.

**Evaluation of other methods that assess genetic similarity between samples**. We considered the following methods:

- HYSIS is intended for tumor-normal concordance verification with a priori knowledge of homozygous germline mutations in the normal tissue[6]. Without considerable modifications, HYSIS is therefore not suitable to handle the general use case that Crosscheck is intended for.
- Bam-matcher is a tool intended for verifying genotype concordance for whole-genome sequencing, whole-exome sequencing, and RNA-sequencing data[7].

Bam-matcher calls programs such as GATK[21] to call variants for each input BAM. Though the resulting variants can be cached to speed up future comparisons, we did not find a way to easily call and store variants for each input bam in the testing set, and without that, performing the hundreds of thousands of benchmarking comparisons becomes unfeasible.

- We did apply the tools Conpair and BAMixChecker to the testing set. Conpair was run with default settings using the standard hg19 SNP panel and the –min-cov parameter set to 1. Pileups were pre-generated using GATK 4.1.7.0 with the recommended settings[8]. BAMixChecker was run with standard settings for hg19 (ref. [9]) and using GATK 4.1.6.0 for variant calling. Conpair outputs a genotype concordance percentage, which should be <50% for different donor and above 80% for same donor datasets. Any genotype concordance between 50 and 80% is considered inconclusive. BAMixChecker outputs a concordance score between 0 and 1 with no explicit inconclusive range. However, we found that BAMixChecker outputs a concordance score of exactly 0 when there is no overlap between the SNP reference panel that the program uses and the input dataset. Therefore, we labeled any result from BAMixChecker with a concordance score of 0 as an inconclusive genotype call. We found that both methods were unable to yield a conclusive result for more than 25% of the comparisons even when the full datasets are considered, and the inconclusive rates became even higher at the lower subsampling rates (Supplementary Fig. 1d). We reasoned that this was likely due to poor overlap between the input datasets and the predefined reference panel of SNPs that both methods use.

**Familial dataset acquisition and processing**. Paired fastqs for RNA-seq data from CEPH/Utah Pedigree 1463 were downloaded from the Gene Expression Omnibus[22] (accession GSE56961). Datasets for the following accessions were downloaded: SRR8505344, SRR8505340, SRR8505343, SRR1258219, SRR1258220, SRR1258218, and SRR8505347. Fastqs were aligned to the GRCh38 reference using STAR[23] 2.6.0c with default parameters. Before analysis, bams were sorted using samtools and duplicate marked/collapsed using Picard's MarkDuplicates. All comparisons were conducted using the default settings and SNP panels for the GRCh38 assembly for each method.

**ENCODE data acquisition**. ENCODE metadata was downloaded from https://www.encodeproject.org/. Metadata were filtered to yield accessions for hg19 ChIP-, RNA-, and DNase-seq ENCODE bams from donors with at least four datasets. These bams were downloaded from a Broad google bucket and processed (see below) with a custom Workflow Description Language[24] script. All dataset accessions and associated metadata are available in Supplementary Data 3.

**ENCODE data processing**. Files were first sorted using *samtools sort*, and filtered using BEDTools[25] in order to only keep reads overlapping SNPs in the haplotype map. This facilitated efficient storage of files, resulting in approximate 10-fold reduction in file size. Finally, duplicates were marked and removed for each file using Picard's MarkDuplicates function with default settings. All comparisons were conducted using the version of CrosscheckFingerprints available in commit #078b0ba of Picard.

**ENCODE genotyping strategy**. To detect mislabeled samples, each dataset is compared against a reference set of three samples that provide a high-quality representation of the "true" genotype for each ENCODE tissue donor. To construct this reference set of samples, a self-LOD score is calculated for each sample by "comparing" each file to itself. This score correlates with the dataset's overlap with the haplotype map, and the highest self-LOD samples are those containing the most genetic information relevant for genotyping. To ensure that the reference set of samples for each tissue donor does not contain any swapped samples, all reference samples are compared against one another to ensure self-consistency, which is defined as an LOD score greater than 5 for all three pairwise comparisons between the three samples. In the case of one swapped sample in this reference set, two negative LOD scores and one positive LOD score will be obtained. In this case, the next highest self-LOD scoring bam replaces the putative swap, and representative concordance is re-assessed. This is repeated until a concordant set is found. More complex patterns of swaps in the representative set are assessed on a case-by-case basis. Finally, all reference samples across all nominal donors are compared against one another in order to identify larger cross-donor swaps and preclude the possibility that all reference samples for a nominal donor are actually swaps from the same true donor.

Each sample not in the reference set is compared against the top three representative samples for its nominal donor. Samples yielding an LOD ≤ −5 against any of the top three representatives are flagged as swaps for review, while those yielding an LOD score between −5 and 5 are flagged as inconclusively genotyped.

**Contamination tests**. Varying numbers of randomly sampled reads from two unrelated ENCODE ChIP-seq datasets, ENCFF005HON ENCFF007DFB, were mixed together to create simulated contaminated datasets. Each mixed sample

consisted of ~5 million reads and contained varying proportions of the original datasets (at intervals of 10%). Mixed samples were then compared to ENCFF007NTA and ENCFF029GAR, which are ChIP-seq datasets from the same donor as ENCFF005HON. Comparisons were conducted on VCF files generated using Picard's ExtractFingerprint utility using Crosscheck's VCF mode with default settings.

**Reporting summary**. Further information on research design is available in the Nature Research Reporting Summary linked to this article.

## Data availability

All data used for benchmarking and ENCODE analysis are available online at https://encodedcc.org/. Specific accessions and relevant metadata for each of the benchmarking experiments are available in Supplementary Data 1 and 2. Accession IDs and metadata for all datasets from ENCODE analysis are available in Supplementary Data 3. Haplotype maps used for benchmarking and ENCODE analysis are available at https://github.com/naumanjaved/fingerprint_maps. RNA-seq data from CEPH/Utah Pedigree 1463 were downloaded from the Gene Expression Omnibus (https://www.ncbi.nlm.nih.gov/geo/) from series GSE56961, using accession IDs: SRR8505344, SRR8505340, SRR8505343, SRR1258219, SRR1258220, SRR1258218, and SRR8505347. 1000 Genome Phase 3 VCFs for hg19 and GRCh38 liftovers were obtained from ftp://ftp-trace.ncbi.nih.gov/1000genomes/ftp. Recombination maps for hg19 and GRCh38 liftovers were obtained from http://bochet.gcc.biostat.washington.edu/beagle/genetic_maps/.

## Code availability

Crosscheck code and documentation is available at https://github.com/broadinstitute/picard. Fingerprint map generation code along with pre-compiled maps and documentation are available at https://github.com/naumanjaved/fingerprint_maps.

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

## Acknowledgements

We would like to thank Jonathan Bloom for his contributions to the LOD score derivation. We thank Liz Gaskell for helpful discussions and comments on the manuscript. N.S. and N.J. are supported by NHGRI UM1HG009390.

## Author contributions

N.J. constructed the haplotype map. Y.F. and T.J.F. designed and wrote CrosscheckFingerprints. N.J., N.S., and Y.F. designed the experimental setup. N.J. and N.S. performed the analyses. C.B.E. and N.S. verified genotyping findings. N.J., Y.F., B.E.B., and N.S. wrote the paper.

## Competing interests

The authors declare no competing interests.
