## [Peer Review File · Nature Communications]

Reviewers' comments:

Reviewer #1 (Remarks to the Author):

In this manuscript, Javed and the authors developed a new algorithm and its implementation Crosscheck that detect sample swaps in NGS cohorts. Unlike several previous tools for the same purpose, Crosscheck uses linkage disequilibrium information to increase accuracy especially in samples with different coverage. In the comparison with NGSCheckMate, the authors showed that Crosscheck can detect sample swaps with a lower error rate in low depth sequencing data (subsampling from original data). Using Crosscheck, followed by manual inspection, the authors detected three types of mislabeling errors in the ENCODE dataset, and reported that about 1% of the data could be mislabeled. Overall, the suggested algorithm adopted new strategies to improve the performance in the given problem, but several concerns remain in the assumption, formality of the method, and practical utility, which should be addressed.

Major comments:

1. It is a good idea to use LD information to increase the coverage of SNPs between two or more samples, especially if SNPs that are covered and genotyped in one sample, but not in other samples. However, SNPs in same LD block are not always in perfect LD. Also SNPs in different LD block are not always independent if their physical distance in chromosome is close. Even though it may not be critical in the actual performance, the description in the introduction is clearly wrong and should be corrected. In the method, the authors used the same simplification including perfect LD in same block and perfect independence in different block. The authors should explicitly clarify their assumption and simplification strategies for implementation, and they are reasonably robust.
2. The formal definition of the likelihood in the Method is a strong point of this manuscript, whereas other tools, to my knowledge, mostly tried to separate mismatches from matches based on heuristic and empirical cutoffs measured from SNP concordance. However, at the later part of the formal quantification, the authors dropped the prior probability term $p(s=0)/p(s=1)$, due to the difficulties in estimation. I understand that the prior is always hard to estimate. But, if prior is not used, the defined posterior odd is not formal anymore, seriously compromising the strong point of the method. Like in other problems such as variant calling, prior probability should be estimated even if it is not perfectly realistic, at least empirically or parameterized for users. And this might affect the final call. For example, let's assume that $p(s=0)$ is 100 times higher than $p(s=1)$ (it is easily assumed that sample swap is a rare event). In that assumption, the final LOD will be shifted toward positive by 2. This means that the current LOD is more favorable for no-swap. Also the 'flagged' and 'inconclusive' samples in the ENCODE analysis can be changed.
3. I have a few more questions about the general utility of the tool:
 - 3-a. Crosscheck showed better performances in low sequencing depth, compared to NGSCheckMate, when subsampled by the factor of <15% of the original data of ENCODE (the actual average read depth is not provided in the manuscript). But, I wonder how frequently sequencing data are generated in this depth. In general WGS, even though the depth is generally low (usually around 30x), the number of SNPs should be more than sufficient due to the large observable area. In typical WES/RNA-seq and targeted panels, sequencing depth is usually much higher (usually >100x). The authors should elaborate more on convincing the practical utility and how frequently Crosscheck can be beneficial in general.
 - 3-b. Crosscheck may be useful in the set of samples, each of which targets disjoint genomic areas such as ChIP-seq and RNA-seq, where conventional tools do not work. Proving this is one way to emphasize the utility.
 - 3-c. The authors provided potential mismatches in the ENCODE dataset. Can these mismatches be found by other tools? If so, how much of them are only detectable by Crosscheck?
4. The author compared the performance only to NGSCheckMate. Is there any specific reason not to include others? There are at least four more tools including Conpair (Bergmann et al, 2016), BAM-matcher (Wang et al, 2016), HYSIS (Schroder et al, 2017), and BamixChecker (Chun et al, 2019). It would be more convincing if the authors provide more comparison results, especially with

recent tools.

5. It would be great to show performance comparisons on more difficult dataset, such as familial data, which have been tested in previous tools such as NGSCheckMate and BamixChecker.

Minor comments:

1. The authors should provide the average read depth after subsampling where Crosscheck starts to outperform other tools.
2. In page 3, Supplementary Fig 1d is not given in the manuscript (probably Supplementary Fig 1c?)

Reviewer #2 (Remarks to the Author):

The authors describe a new method/tool called Crosscheck that is able to assess sample relatedness from NGS data by using a Bayesian approach applied to LD-blocks. The tool is widely available as part of the Picard suite and can be readily used to find identity mismatches between different sets of NGS data. This approach generates an LOD score, and using a subset of the data from the ENCODE project the authors find different useful thresholds which are then used on the entire ENCODE dataset, finding about 1% of the samples have potential samples swaps/mismatches that are then studied in detail. The model used is statistically sound and represents a novel and useful approach to the problem of sample swapping and the manuscript merits to be published contingent on addressing some concerns, which I am including below:

Major concerns:

1) In page 3), second paragraph, the authors used a simplified approach (that dramatically saves time) to test their approach on the ENCODE dataset by "comparing each dataset to a set of three representative datasets from its annotated donor". This strategy seems sound, however I believe the choice of the number of representative datasets (three) could have been tested using the dataset where they did the pilot (279 sets from page 2 where they perform all to all comparisons). Using this pilot dataset they can show that their strategy of just using 3 representatives does not alter significantly FMR and/or FMR while saving a lot of computation time.

2) On page 3, on the last paragraph, they identify some mismatched-cases where cell types are derived from multiple donors. How does the LOD score looks like in those cases and are those score distribution different that just simple sample swaps? Is the LOD score affected by the proportion of a particular sample involved in the mixture?

3) In the methods section the authors refer to a report that outlines in more detail the statistical approach followed. Such report is authored by Jon Bloom and Yossi Farjoun. Yossi is included in the list of authors in this manuscript, however Jon Bloom is not. Is there a reason for that? It seems the manuscript benefits greatly from such method/approach so it would seem natural to include both of the authors.

Minor comment:

4) Although not the main objective of this manuscript, the authors perhaps should comment on how this method is affected by sample coming from related individuals. Also perhaps some comments on what happens when sample contamination of other samples occur.

Detailed responses to reviewer's comments

In the following pages, text in black is the original comment by the reviewers and our response is in purple.

Reviewer 1:

In this manuscript, Javed and the authors developed a new algorithm and its implementation Crosscheck that detect sample swaps in NGS cohorts. Unlike several previous tools for the same purpose, Crosscheck uses linkage disequilibrium information to increase accuracy especially in samples with different coverage. In the comparison with NGSCheckMate, the authors showed that Crosscheck can detect sample swaps with a lower error rate in low depth sequencing data (subsampling from original data). Using Crosscheck, followed by manual inspection, the authors detected three types of mislabeling errors in the ENCODE dataset, and reported that about 1% of the data could be mislabeled. Overall, the suggested algorithm adopted new strategies to improve the performance in the given problem, but several concerns remain in the assumption, formality of the method, and practical utility, which should be addressed.

Major comments:

1. It is a good idea to use LD information to increase the coverage of SNPs between two or more samples, especially if SNPs that are covered and genotyped in one sample, but not in other samples. However, SNPs in same LD block are not always in perfect LD. Also SNPs in different LD block are not always independent if their physical distance in chromosome is close. Even though it may not be critical in the actual performance, the description in the introduction is clearly wrong and should be corrected. In the method, the authors used the same simplification including perfect LD in same block and perfect independence in different block. The authors should explicitly clarify their assumption and simplification strategies for implementation, and they are reasonably robust.
 - We completely agree that the approximations that the method uses must be stated as clearly and explicitly as possible. We have revised and added text and emphases in both the main text and the Methods section to make the assumptions clearer.
 - In addition, to demonstrate the robustness of the method to the specific choices of 0.85 and 0.1 for the intra- and inter-block correlations cutoffs respectively, we have generated LD blocks maps with more relaxed (0.80, 0.15) and more stringent (0.90, 0.05) cutoffs. When applied to the test set, the (extremely low) error rates were unaffected by these changes to the cutoff parameters. A sentence to that effect has also been added to the manuscript.
2. The formal definition of the likelihood in the Method is a strong point of this manuscript, whereas other tools, to my knowledge, mostly tried to separate mismatches from matches based on heuristic and empirical cutoffs measured from SNP concordance. However, at the later part of the formal quantification, the authors dropped the prior probability term $p(s=0)/p(s=1)$, due to the difficulties in estimation. I understand that the prior is always hard to estimate. But, if prior is not used, the defined posterior odd is not formal anymore, seriously compromising the strong point of the method. Like in other problems such as variant calling, prior probability should be estimated even if it is not perfectly

realistic, at least empirically or parameterized for users. And this might affect the final call. For example, let's assume that $p(s=0)$ is 100 times higher than $p(s=1)$ (it is easily assumed that sample swap is a rare event). In that assumption, the final LOD will be shifted toward positive by 2. This means that the current LOD is more favorable for no-swap. Also the 'flagged' and 'inconclusive' samples in the ENCODE analysis can be changed.

- It is important to explicitly include the prior odds in the LOD calculation despite it being difficult to estimate. We have corrected the methods section to reflect that the prior odds are not carried through into the LOD calculation because we conservatively assume that $p(s=0)/p(s=1) = 1$. This conservative prior is suitable for genotyping consortium projects such as ENCODE where it is critical to maintain swap detection sensitivity. In other use cases, different prior odds can be incorporated by simply shifting the final LOD score by $\log_{10}(p(s=0)/p(s=1))$. Users may thus adjust the LOD score in cases where a less conservative prior might be more appropriate.

3. I have a few more questions about the general utility of the tool:

- a. Crosscheck showed better performances in low sequencing depth, compared to NGSCheckMate, when subsampled by the factor of <15% of the original data of ENCODE (the actual average read depth is not provided in the manuscript). But, I wonder how frequently sequencing data are generated in this depth. In general WGS, even though the depth is generally low (usually around 30x), the number of SNPs should be more than sufficient due to the large observable area. In typical WES/RNA-seq and targeted panels, sequencing depth is usually much higher (usually >100x). The authors should elaborate more on convincing the practical utility and how frequently Crosscheck can be beneficial in general.
 - We agree that it is important to highlight the applications where Crosscheck's superior performance at low read-depths are particularly important. We have revised the text to highlight several examples where shallow sequencing is utilized in NGS-experiments: (1) experiments where many different samples and assays are multiplexed and sequenced in the same sequencing run, (2) low-depth WGS/WES sequencing increasingly utilized in large-scale population based sequencing efforts such as 1000 Genomes (3) samples which are added to sequencing runs as "spike-ins" in order to preserve the original sample or verify that the library preparation was successful, (4) cancer genomics applications which utilize low depth sequencing to characterize large-structural variants. We also acknowledge the value of providing the actual average read count for each of the down-sampling percentages shown in the benchmarking experiments. We direct the reviewer's attention to Supplementary Fig. 1B, showing the distribution of read counts for the different assay types at each of the down-sampling percentages.
- b. Crosscheck may be useful in the set of samples, each of which targets disjoint genomic areas such as ChIP-seq and RNA-seq, where conventional tools do not work. Proving this is one way to emphasize the utility.
 - We thank the reviewer for prompting us to highlight this important point. We have focused on the non-overlapping marks of H3K27me3 and H3K27ac, and added a supplementary plot (Supplementary Fig. 1e) to demonstrate that in the subset of test-set comparisons involving these marks (subsampled at 5%):

- NGSCheckmate has a significantly higher error rate when comparing K27me3-K27ac datasets than when comparing within each target type.
 - Crosscheck is still able to correctly call all comparisons without errors.
 - We also added in the main text a short description of a specific pair of datasets (again K27me3 and K27ac) that overlap at 0.02% of the genome, where NGSCheckmate fails to distinguish the source donors, but Crosscheck flags them as a potential mismatch.
- c. The authors provided potential mismatches in the ENCODE dataset. Can these mismatches be found by other tools? If so, how much of them are only detectable by Crosscheck?
- It is possible that the mismatches in ENCODE could be detected by other tools. Beyond demonstrating Crosscheck's superiority in performance when coverage is low or when datasets hardly overlap, we do not make the claim that it is the only tool that could be used to detect mismatches in ENCODE. That said, in our experience we saw that other tools weren't effective at comparing such diverse data types. Alongside the benchmarking experiments, the purpose of the analysis of ENCODE data is to further demonstrate the utility of Crosscheck and its scalability, and to provide the community and the consortium with valuable data QC for this highly utilized data repository.
4. The author compared the performance only to NGSCheckMate. Is there any specific reason not to include others? There are at least four more tools including Conpair (Bergmann et al, 2016), BAM-matcher (Wang et al, 2016), HYSIS (Schroder et al, 2017), and BamixChecker (Chun et al, 2019). It would be more convincing if the authors provide more comparison results, especially with recent tools.
- Comparison to existing tools is of course critical when introducing a new one, and we thank the reviewer for prompting us to revisit this more thoroughly. In examining possible methods to compare Crosscheck to we considered the following criteria:
 - Straightforward deployment of the tool
 - Applicability or straightforward adaptability to a variety of data types and use cases
 - Scalability and cloud portability that makes processing a large number of comparisons feasible
 - Following the reviewer's suggestion, and after additional literature search, we examined (in addition to NGSCheckmate), Conpair (Bergmann et. al 2016), BamixChecker (Chun et. al 2019), HYSIS, and Bam-matcher.
 - HYSIS is intended for tumor-normal concordance verification with a priori knowledge of homozygous germline mutations in the normal tissue. Without considerable modifications, HYSIS is therefore not suitable to handle the general use case that Crosscheck is intended for.
 - There is no clear method for running Bam-matcher in batch-mode, which would be required for the hundreds of thousands of pairwise comparisons conducted during benchmarking.

- We did apply Conpair and BamixChecker to the test set. We found that both methods were unable to yield a conclusive result for more than 25%(!) of the comparisons even when the full datasets are considered, and the inconclusive rates became even higher at the lower subsampling rates. The reason for this is that both methods rely on predefined reference panels of SNPs for comparing datasets. The large number of inconclusive comparisons was the result of testing datasets frequently not having sufficient overlap with these reference panels to allow meaningful comparisons.
 - We added a description of these results in the main text of the manuscript, and a supplementary figure.
5. It would be great to show performance comparisons on more difficult dataset, such as familial data, which have been tested in previous tools such as NGSCheckMate and BamixChecker.
- We thank the reviewer for this suggestion and have incorporated an analysis of familial data from the CEPH/Utah Pedigree 1463. We obtained RNA-seq data from 7 individuals in this pedigree from the Gene Expression Omnibus (accession GSE56961). We classified all 21 pairwise comparisons using Crosscheck, NGSC, Conpair, and BamIXChecker. While Crosscheck perfectly differentiated between all related individuals, NGSC successfully classified only 57% of pairwise comparisons successfully. Similar to the test set, BamixChecker and Conpair yielded inconclusive results for many of these comparisons. We refer to this analysis in the main text, and added text and a figure to the supplementary material.

Minor comments:

1. The authors should provide the average read depth after subsampling where Crosscheck starts to outperform other tools.
 - We provide the average *total number of reads* after subsampling where Crosscheck outperforms other tools. For most of the data types, providing an average *read depth* is not informative since the assays target specific genomic regions (i.e. accessible peaks or specific histone marks). The distribution of the total number of reads for each type of assay is provided in Supplementary Fig. 1B.
2. In page 3, Supplementary Fig 1d is not given in the manuscript (probably Supplementary Fig 1c?)
 - We have corrected this typo.

Reviewer 2:

The authors describe a new method/tool called Crosscheck that is able to assess sample relatedness from NGS data by using a Bayesian approach applied to LD-blocks. The tool is widely available as part of the Picard suite and can be readily used to find identity mismatches between different sets of NGS data. This approach generates an LOD score, and using a subset of the data from the ENCODE project the authors find different useful thresholds which are then used on the entire ENCODE dataset, finding about 1% of the samples have potential samples swaps/mismatches that are then studied in detail. The model used is statistically sound and represents a novel and useful approach to the problem of sample swapping and the manuscript merits to be published contingent on addressing some concerns, which I am including below:

Major concerns:

1. In page 3), second paragraph, the authors used a simplified approach (that dramatically saves time) to test their approach on the ENCODE dataset by "comparing each dataset to a set of three representative datasets from its annotated donor". This strategy seems sound, however I believe the choice of the number of representative datasets (three) could have been tested using the dataset where they did the pilot (279 sets from page 2 where they perform all to all comparisons). Using this pilot dataset they can show that their strategy of just using 3 representatives does not alter significantly FMR and/or FMR while saving a lot of computation time.
 - As the reviewer implies, in the testing set nearly all the datasets from a donor yielded positive LODs when compared to any other dataset from the same donor, and negative LODs when compared to any other dataset from other donors. This implies that the match/flagged call would be the same whether we compared a query dataset to a single representative from its nominal donor or to all available datasets from that donor.
 - It is thus not surprising that in the examination of ENCODE data, in the overwhelming majority of cases, datasets had either positive LOD with all 3 representatives of a donor, or 3 negative ones. We thus saw no indication in the data that using a single representative from each donor would have missed any flagged datasets that were captured by using 3 representatives. This number of representatives is thus somewhat arbitrary and possibly overly conservative.
 - We draw the reviewer's attention to the "ENCODE genotyping strategy" section of the Methods, where the selection of representative datasets is explained in more detail.
2. On page 3, on the last paragraph, they identify some mismatched-cases where cell types are derived from multiple donors. How does the LOD score looks like in those cases and are those score distribution different that just simple sample swaps? Is the LOD score affected by the proportion of a particular sample involved in the mixture?
 - This comment spurred us to examine in-silico the effects of a mixture of cells, as may happen when there is cross-sample contamination. This analysis is described in the response to the last comment of this reviewer.
 - To be clear, the specific case that the reviewer refers to does not involve the mixture of different cells from different donors in any single sample. Rather than a mixture, it is the fact that over time, different batches of cells (supplied by a third party vendor), all nominally attributed to "the donor of HUVEC cells", have actually been derived from several distinct individuals. From the genetic perspective, the fact that these happened to be of the same cell type is inconsequential. The LOD split between positive and negative LODs was the same as when comparing datasets from any donors.
3. In the methods section the authors refer to a report that outlines in more detail the statistical approach followed. Such report is authored by Jon Bloom and Yossi Farjoun. Yossi is included in the list of authors in this manuscript, however Jon Bloom is not. Is there a reason for that? It seems the manuscript benefits greatly from such method/approach so it would seem natural to include both of the authors.

- Jon Bloom has been aware of the writing of this manuscript. We have reached out to him again and he has been added to the acknowledgements paragraph.

Minor comment:

4. Although not the main objective of this manuscript, the authors perhaps should comment on how this method is affected by sample coming from related individuals. Also perhaps some comments on what happens when sample contamination of other samples occur.
- We thank the reviewer for this suggestion and have incorporated an analysis of familial data from the CEPH/Utah Pedigree 1463. We obtained RNA-seq data from 7 individuals in this pedigree from the Gene Expression Omnibus (accession GSE56961). We classified all 21 pairwise comparisons using Crosscheck, NGSC, Conpair, and BamIXChecker. While Crosscheck perfectly differentiated between all related individuals, NGSC successfully classified only 57% of pairwise comparisons successfully. Similar to the test set, BamixChecker and Conpair yielded inconclusive results for many of these comparisons. We mention this analysis in the main text, and added text and a figure to the supplementary material.
 - We have also added an analysis of the patterns of Crosscheck LOD scores that are expected in the case of sample contamination resulting in a mixture of cells from more than one donor. In this analysis we created in-silico mixtures of datasets from two donors at different proportions, and compared them to two other datasets from one of the donors. The analysis showed that there exists a region in which a mixed sample would seem to match one dataset from a donor, but to disagree with another dataset from the same donor. Such inconsistent pattern was observed on a small fraction (0.4%) of ENCODE datasets that we tested, and this analysis suggests a possible explanation for that pattern. This analysis is mentioned in the main text and explained in the Methods section and in Supplementary Fig. 3.

REVIEWERS' COMMENTS:

Reviewer #1 (Remarks to the Author):

The authors successfully address most of my previous concerns. I appreciate the authors' efforts. I think the manuscript is now publishable. I only have a few minor questions and comments regarding this revision.

1. In the comparison with other tools, I agree with the authors that HYSIS and Bam-matcher are difficult to apply. For BamixChecker, however, users can provide BED files to specify the target sites. I wonder if the authors applied all the appropriate parameters for the comparison.
2. I found that the familial dataset has been mapped to hg38, but the authors mentioned that all settings were based on hg19.

Reviewer #2 (Remarks to the Author):

The authors have addressed all of my comments and have come out with interesting analysis addressing familial relatedness and sample swapping that were not included in the original manuscript which have enhanced the paper and made its point much stronger. I believe the paper merits publication.

Detailed responses to reviewer's comments

Reviewer 1:

The authors successfully address most of my previous concerns. I appreciate the authors' efforts. I think the manuscript is now publishable. I only have a few minor questions and comments regarding this revision.

1. In the comparison with other tools, I agree with the authors that HYSIS and Bam-matcher are difficult to apply. For BamixChecker, however, users can provide BED files to specify the target sites. I wonder if the authors applied all the appropriate parameters for the comparison.

It is indeed the case that BamixChecker can be used with a BED file. This, however, is designed for targeted assays, where only a small predefined subset of the genome is interrogated. None of the NGS data that we discuss and analyze in the manuscript is of that type, and we therefore used BamixChecker in its recommended default mode which uses a reference panel that is provided with the tool. We consider it outside the scope of this manuscript to examine the performance of BamixChecker in conjunction with the haplotype map that we have generated.

2. I found that the familial dataset has been mapped to hg38, but the authors mentioned that all settings were based on hg19.

We have modified the text in the methods section "Familial dataset acquisition and processing" to indicate that all familial comparisons were conducted using the recommended settings/panels for GRCh38 for each tool.

Reviewer 2:

The authors have addressed all of my comments and have come out with interesting analysis addressing familial relatedness and sample swapping that were not included in the original manuscript which have enhanced the paper and made its point much stronger. I believe the paper merits publication.